# Aloe-Emodin Ameliorates Cecal Ligation and Puncture-Induced Sepsis

**DOI:** 10.3390/ijms241511972

**Published:** 2023-07-26

**Authors:** Jingqian Su, Siyuan Chen, Jianbin Xiao, Zhihua Feng, Shan Hu, Qiaofen Su, Qi Chen, Duo Chen

**Affiliations:** 1Key Laboratory of Innate Immune Biology, Biomedical Research Center of South China, College of Life Science, Fujian Normal University, Fuzhou 350117, China; sjq027@fjnu.edu.cn (J.S.); fzhfjnu@163.com (Z.F.); qbx20220152@yjs.fjnu.edu.cn (S.H.); 15260328242@163.com (Q.S.); 2Provincial University Key Laboratory of Microbial Pathogenesis and Interventions, College of Life Sciences, Fujian Normal University, Fuzhou 350117, China; 3The Public Service Platform for Industrialization Development Technology of Marine Biological Medicine and Products of the State Oceanic Administration, Fujian Key Laboratory of Special Marine Bioresource Sustainable Utilization, Southern Institute of Oceanography, College of Life Science, Fujian Normal University, Fuzhou 350117, China; histxinkexueyuan@163.com (S.C.); xiaojianbin0802@163.com (J.X.)

**Keywords:** sepsis, aloe-emodin, inflammation, cecal ligation puncture, gut microbiota

## Abstract

Sepsis remains a major challenge owing to its severe adverse effects and high mortality, against which specific pharmacological interventions with high efficacy are limited. Mitigation of hyperactive inflammatory responses is a key factor in enhancing the likelihood of survival in patients with sepsis. The *Aloe* genus has several health benefits, including anti-inflammatory properties. The toxicological implications of aloe-emodin (AE), extracted from various *Aloe* species, remain uncertain in clinical contexts. However, AE has been shown to inhibit inflammatory responses in lipopolysaccharide-induced mice, indicating its potential as a therapeutic approach for sepsis treatment. Nonetheless, there is a paucity of data regarding the therapeutic benefits of AE in the widely recognized cecal ligation and puncture (CLP)-induced sepsis model, which is commonly used as the gold standard model for sepsis research. This study demonstrates the potential benefits of AE in the treatment of CLP-induced sepsis and investigates its underlying mechanism, along with the efficacy of postoperative AE treatment in mice with CLP-induced sepsis. The results of this study suggest that AE can mitigate sepsis in mice by diminishing systemic inflammation and regulating the gut microbiota. The study provides novel insights into the molecular mechanisms underlying the anti-inflammatory effects of AE.

## 1. Introduction

Sepsis is a life-threatening condition induced by infections that result from an imbalance between inflammation and immunosuppression [1]. Currently, the most recent scholarly investigation on sepsis is denoted as “Sepsis 3.0”. Within the framework of “Sepsis 3.0”, sepsis is delineated as a perilous impairment of organ function, arising from an aberrant host reaction to infection. This implies that the body’s defensive response to infection detrimentally affects its own tissues and organs, thereby posing a threat to life [2]. Sepsis remains a significant public health concern that affects approximately 18 million individuals annually worldwide and has a mortality rate ranging from 28 to 50%. In patients with coronavirus disease (COVID-19) who required hospitalization in intensive care units, approximately 78% showed clinical manifestations of sepsis-3. To date, no targeted pharmacotherapies are available for the treatment of sepsis [3,4]. The primary therapeutic approach involves the mitigation of excessive inflammation during infection [5]. The elimination of pro-inflammatory cytokines can effectively reduce the incidence of inflammation-induced organ damage and mortality [6].

Three primary animal models of sepsis have been established, namely cecal ligation and puncture (CLP), *Escherichia coli* infusion, and lipopolysaccharide (LPS) infusion. The CLP model, owing to its high degree of realism, is commonly used as the gold standard for sepsis research [7]. Based on this model, the introduction of intestinal bacteria, fungi, and metabolites into the abdominal cavity can result in abdominal infection and subsequent systemic septicemia [8]. The imbalance in the intestinal flora associated with sepsis is linked to organ inflammation; nevertheless, the precise mechanism by which alterations in intestinal flora affect sepsis pathogenesis remains unclear [9].

The *Aloe* genus has numerous health benefits, including anti-inflammatory, emollient, gastroprotective, purgative, antimicrobial, anticancer, aphrodisiac, antifungal, hypoglycemic, and antioxidant properties [10]. Aloin (Alo; CAS:1415-73-2) and aloe-emodin (AE; CAS:481-72-1), natural anthraquinone compounds, have been extracted from various *Aloe* species [11] (Figure 1). We have previously demonstrated the efficacy of Alo in activity restoration and appetite loss, as well as in increasing the survival rate of mice with CLP-induced sepsis (data unpublished).

Over the past decade, numerous reports have provided evidence of the hepatotoxic and nephrotoxic properties of AE [12]. Currently, the European Union has imposed a ban on the utilization of AE due to the potential genotoxicity of its derivatives [13]. Nevertheless, the results obtained from an in vivo comet test have unequivocally established that aloe-emodin does not exhibit genotoxic characteristics [14]. Hence, the precise clinical implications of these toxic effects remain uncertain [15]. In contrast to food, AE, being a potential therapeutic agent, should be permitted to induce specific adverse reactions in the human body. However, it is imperative to conduct further investigations into the pharmacological side effects of AE.

In the intestinal tract, Alo hydrolyzes one glucose molecule into AE, which stimulates the intestinal wall, enhances intestinal peristalsis, and promotes the expulsion of intestinal waste [16]. Hu et al. provided evidence that AE exerts anti-inflammatory effects by inhibiting the production of inflammatory factors in RAW264.7 macrophages induced by LPS [17]. Similarly, Gao et al. demonstrated the protective effects of AE against LPS-induced inflammation in a mouse model. These findings suggest AE as a viable therapeutic agent against sepsis [18].

Nonetheless, data on the therapeutic effect of AE in a CLP-induced sepsis model are lacking. Therefore, this study examines, to the best of our knowledge for the first time, the therapeutic efficacy of AE in a mouse model of CLP-induced sepsis to present evidence of the advantageous effect of AE on sepsis and elucidate the underlying molecular mechanisms. This study involved the implementation of postoperative AE therapy in a CLP-induced sepsis model. The principal objective was to explore the anti-inflammatory impact, assess modifications in immune cells, and scrutinize the composition of intestinal flora. The outcomes of this research are expected to contribute to the advancement of AE as a potential therapeutic approach for sepsis.

## 2. Results

### 2.1. AE Modulated Inflammatory Cytokine Expression on LPS-Stimulated RAW264.7 Cells

The cytotoxicity of AE (0–300 μmol/L) on RAW264.7 cells was negligible (Figure 2A); indeed, the combination of AE (300 mol/L) and LPS (0.1 mg/L) caused no observable cytotoxicity (Figure 2B). Furthermore, AE (300 mol/L) inhibited the LPS-induced upregulation of IL-6, IL-1β, and TNF-α induced by LPS in RAW264.7 cells (*p* < 0.05; Figure 2C–E). Subsequent enzyme-linked immunosorbent assays (ELISA) confirmed the quantitative reverse transcription polymerase chain reaction (RT-qPCR) results (*p* < 0.05; Figure 2F–H). These results indicate that AE markedly inhibited the upregulation of inflammatory cytokines in LPS-stimulated RAW264.7 cells.

### 2.2. Therapeutic Effect of AE in Mice with CLP-Induced Sepsis

The effects of AE (0.1–10 mg/kg) were evaluated in the mouse model of CLP-induced sepsis (Figure 3A). The AE (10 mg/kg)-treated group showed reduced agglomeration, increased activity, and improved appetite compared with the untreated CLP group (Figure 3B). The murine sepsis score (MSS) induced by CLP was reduced following AE treatment (*p* < 0.01, Figure 3C). The survival rate of the AE (10 mg/kg) was higher than that of the CLP-treated group (50% vs. 20%, *p* < 0.05; Figure 3D). These findings suggest that AE plays a critical role in the treatment of mice with CLP-induced sepsis.

### 2.3. AE Reduced Inflammatory Cytokine Levels in Mice with CLP-Induced Sepsis

To determine whether AE modulates inflammatory cytokine expression in vivo, we examined whether AE inhibits CLP-induced inflammatory cytokine levels in the serum, lungs, liver, and heart. AE treatment reduced the CLP-induced increase in IL-1β, IL-6, and TNF-β levels in the serum (*p* < 0.01, Figure 4A–C). Similar results were obtained for the lung (Figure 4D–F), liver (Figure 4G–I), and heart (Figure 4J–L) tissues (*p* < 0.05). Following CLP induction, RT-qPCR was used to examine the effect of AE on inflammatory cytokines (*IL-6*, *IL-1β*, and *TNF-α*) in the lung (Figure 5A–C), liver (Figure 5D–F), and heart (Figure 5G–I) tissue. A significant reduction was observed in inflammatory cytokine expression in the AE-treated group, compared with the CLP-treated group (*p* < 0.05, Figure 5). These results indicate that AE markedly inhibited inflammatory cytokine production in mice with CLP-induced sepsis.

### 2.4. AE Ameliorated CLP-Induced Pathology

To determine whether AE ameliorated the CLP-induced pathological changes in the lung, liver, and heart tissue, hematoxylin and eosin (H&E) staining was performed 12 h after AE treatment. AE treatment alleviated lung injury and attenuated alveolar wall thickening and hemorrhage induced by CLP (*p* < 0.05; Figure 6A–D). A similar trend was observed in the liver, with the AE-treated group presenting a lower level of liver injury, reduced inflammatory cells, decreased swelling, and a more regular arrangement of hepatocytes than the CLP group (*p* < 0.05; Figure 6E–H). Following treatment with AE, mice in the AE-treated group showed reduced cardiomyocyte spaces and an improvement in myocardial histopathological damage (*p* < 0.05; Figure 6I–L). The extent of histopathological damage was assessed using pathological scores (Figure 6M–O). These findings suggest that AE substantially ameliorated CLP-induced damage to the respiratory, hepatic, and cardiovascular systems of mice.

### 2.5. AE Altered the Hematology Parameters in Mice with CLP-Induced Sepsis

To determine whether AE improved the generation and proliferation of CLP-induced immune cells, we analyzed hematological parameters 12 h after AE treatment. AE treatment reduced the total number of white blood cells, lymphocytes, and granulocytes in the blood of CLP-treated mice (*p* < 0.05, Figure 7A–C). A similar trend was observed in the bloodstream; monocytes and hematocrit were decreased in mice 12 h after AE treatment compared with those in the CLP group (*p* < 0.05, Figure 7E–H). However, no change was observed in red blood cell count (*p* > 0.05, Figure 7D). The platelet crit level in the AE-treated group was higher than that in the CLP group (*p* < 0.05, Figure 7I). Administration of AE restored the generation and proliferation of immune cells near normal levels in mice with CLP-induced sepsis.

### 2.6. AE Regulated Intestinal Flora Homeostasis in Mice with CLP-Induced Sepsis

Intestinal flora was sequenced and analyzed using metagenomic methods to explore whether AE improved CLP-induced intestinal flora homeostasis in septic mice. According to the Venn diagram of gut microorganisms, AE reduced the number of intestinal bacteria-specific flora in mice with sepsis, restored intestinal flora homeostasis, and reduced inflammatory responses (Figure 8A). Principal coordinates analysis (PCoA) indicated differences in the gut microbial communities among the four groups (*p* < 0.05, Figure 8B). No notable changes in the ACE, Chao1, Shannon, and Simpson indices were observed in the AE treatment group compared with those in the CLP group (*p >* 0.05, Figure 8C–G). These findings suggest that AE had no marked effect on the richness and diversity of the gut microbiota in septic mice.

To conduct a more comprehensive examination of the diversity of the microbiota structure, the relative abundance at the phylum, family, and genus levels was analyzed (Figure 8H,I and A,B). AE reduced *Akkermansia* in mice with CLP-induced sepsis (*p* < 0.05, Figure 8J), while no significant alterations were observed in the levels of *Dubosiella*, *Muribaculaceae*, and *Prevotellaceae_UCG_001* (*p >* 0.05, Figure 8K–M).

The microbial groups were also analyzed using a linear discriminant analysis effect size (LEfSe) assay (Figure 9C). Using the screening criterion of linear discriminant analysis (LDA) values > 4, 50 distinct species were identified (Figure 9D).

AE can modulate the prevalence of select microbiota, diminish the population of pro-inflammatory pathogenic bacteria, and elicit favorable regulation of the intestinal microbiota in septic mice.

### 2.7. AE Facilitated the Production of SCFAs in Mice Afflicted with CLP-Induced Sepsis

Short-chain fatty acids (SCFAs), important metabolites of the intestinal microbiota, are essential for regulating inflammation and immunity. Following AE treatment, the levels of acetic, propionic, butyric, and isovaleric acids were higher in the AE-treated than the CLP groups (*p* < 0.05, Figure 10A–D). Subsequently, correlation analysis was performed to investigate the relationship between pro-inflammatory cytokines, SCFAs, and the composition of the intestinal microflora. At the genus level, *Parasutterella*, *Escherichia,* and *Lachnospiraceae_NK4A136_group* were positively correlated with TNF-α and IL-1β levels in mouse serum (*p* < 0.05, Figure 10E). *Prevotellaceae_UCG-001*, *Eubacterium*, *Clostridia_UCG_014*, *Alloprevotella,* and *Dubosiella* were negatively correlated with TNF-α, IL-6, and IL-1β in mouse serum (*p* < 0.05, Figure 10E). Correlation analysis of SCFAs, *Eubacterium fissicatena*, *Clostridia_UCG_014*, *Alloprevotella,* and *Dubosiella* showed negative association with isobutyric, acetic, propionic, and butyric acids (*p* < 0.05, Figure 10E). *Escherichia*, *Lachnospiraceae_NK4A136_group*, and uncultured *Bacteroidales_bacterium* were positively correlated with the levels of isobutyric, acetic, propionic, and butyric acids (*p* < 0.05, Figure 10E). The levels of acetic, propionic, butyric, and isobutyric acids were positively correlated with the abundance of *Alloprevotella*, *Dubosiella, Mucispirillum, Prevotellaceae_UCG-001*, *Eubacterium*, *Clostridia_UCG_014*, and *Lactobacillus* (Figure 10F).

Therefore, the mechanism of AE activity may involve the production of SCFAs by the intestinal microbiota, thereby aiding in the regulation of sepsis by enhancing the beneficial acid ratio.

### 2.8. AE Inhibited the CLP-Induced Inflammatory Response

To explore the mechanisms by which AE impedes the inflammatory response triggered by CLP, we assessed the relevant markers of inflammatory response activation using immunoblotting and immunohistochemical analysis of lung tissue. CLP treatment activated the NF-κB signaling pathway, as evidenced by the marked elevation in the phosphorylation of both NF-κB (p65) and IKBα; conversely, AE treatment reduced these changes (*p* < 0.05; Figure 11A–C). AE also suppressed the phosphorylation of NF-κB (p65) in the lung tissue of mice with CLP-induced sepsis (Figure 11D–G). These findings suggest that AE exerts a protective effect against CLP-induced sepsis through inhibition of the NF-κB signaling pathway.

## 3. Discussion

To the best of our knowledge, this is the first study on the therapeutic efficacy of AE in CLP-induced sepsis in mice. Our findings suggest that AE possesses anti-inflammatory properties. Specifically, the administration of 10 mg/kg/day of AE significantly increased the survival rate from 20 to 50% in CLP-induced septic mice. The results of the histopathological analyses indicated that AE effectively mitigated lung, liver, and heart damage induced by CLP in mice. Additionally, AE substantially inhibited the excessive activation of immune cells in the peripheral blood of mice with CLP-induced sepsis. Following AE treatment, mice with CLP-induced sepsis demonstrated a notable prevalence of probiotics and a diminished prevalence of pathogens or opportunistic pathogens. In contrast, AE substantially increased acetic, propionic, butyric, and isovaleric acid levels.

Reduced expression of inflammation-related factors was highly correlated with the abundance of *Prevotellaceae_UCG-001, Eubacterium, Clostridia_UCG_014, Alloprevotella,* and *Dubosiella.* Moreover, AE inhibited IκBα phosphorylation induced by CLP.

In the context of sepsis, the secretion of inflammatory cytokines by monocytes and neutrophils, along with the promotion of lymphocyte proliferation, results in the development of a “cytokine storm” that leads to cytotoxicity [19]. Therefore, the management of inflammation is a crucial aspect of sepsis treatment [20]. AE pharmacological activity is closely linked to its chemical structure. The anthraquinone ring and two phenolic hydroxyl groups of AE influence its anti-inflammatory and antimicrobial properties [21,22]. AE exerted a discernible inhibitory effect on *Acinetobacter baumannii*, *Candida albicans*, and *Staphylococcus aureus* [23].

AE also exerted an anti-inflammatory effect by inhibiting the LPS-induced Toll-like receptor 2 (TLR2)-mediated NF-κB and MAPK signaling pathways in macrophages through a reduction in the mRNA expression levels of iNOS, IL-6, and IL-1β [17,18,24]. Additionally, AE decreased the release of HMGB1 by restoring the expression of the endothelial association protein, ZO-1/2, ultimately inhibiting the formation and activation of the NLRP3 inflammasome regulated by NLRP1 ubiquitination and consequently reducing the inflammatory response [12]. AE can impede apoptosis by reducing the cysteine aspartate protease (Caspase-3) expression levels [25] and can treat subarachnoid hemorrhage (SAH) by modulating the NF-κB and cyclic adenosine phosphate/protein kinase A/responsive element binding protein pathways, thereby inhibiting hemorrhage-induced inflammatory cytokines and nerve cell apoptosis [26]. These findings indicate that AE merits further investigation as a means of preventing and managing sepsis.

Healthy organisms maintain homeostasis by balancing the host inflammatory response and gut-mediated immunity in a dynamic equilibrium [26]. Sepsis-induced alterations in the composition of the gut microbiome lead to organ dysfunction through an intricate interplay between the microbiota and the immune system. Disturbances in homeostasis are manifested as shifts in bacterial abundance, gut bacterial translocation, and dysbiosis [27].

In the current study, the composition of intestinal microorganisms in mice with CLP-induced sepsis changed following AE treatment, which was related to the expression of inflammatory factors. Hence, the correlation between pro-inflammatory factors and the intestinal microbiota was further analyzed.

For the first time, we demonstrated that AE treatment markedly increased the abundance of *Dubosiella*, *Muribaculaceae*, *Alloprevotella*, *Prevotellaceae_UCG-001*, *Eubacterium*, and *Clostridia_UCG_014* and decreased that of *Akkermansia*, *Bacillus*, *Klebsiella*, *Morganella*, and *Streptococcus*. Spearman correlation analysis showed that *Clostridia_UCG_014* correlated negatively with IL-1β, IL-6, and TNF-α expression induced by CLP and positively with the generation of acetic and propionic acids.

*Akkermansia* has beneficial effects on host metabolism. The cell membrane of *Akkermansia* is characterized by the presence of a lipid molecule, diacylphosphatidyl ethanolamine, with two branched chains (a15:0-i15:0 PE), which constitute the predominant component of the *Akkermansia* lipid membrane, accounting for approximately 50% of its composition. This lipid molecule has shown a dose–response association in the stimulation of TNF-α and IL-6 [28,29].

*Clostridia_UCG_014* can withstand gastric acid upon entering the intestine, while concurrently facilitating the proliferation of advantageous bacteria, suppressing the growth of detrimental intestinal microorganisms, reinstating intestinal flora, enhancing immunity, and promoting the assimilation, digestion, and absorption of nutrients. This genus confers numerous benefits to the host, including reduction in inflammation, regulation of immunity, attenuation of carbon metabolism, and secretion of SCFAs [30,31,32].

Indigestible polysaccharides (e.g., dietary fibers) fermented under anaerobic conditions produce SCFAs. Less than 10% of SCFAs are excreted in the stool following production in the gastrointestinal tract, whereas the remaining SCFAs are absorbed to provide energy to the host epithelium, thereby regulating the immune system and maintaining homeostasis. SCFAs are transported to other organs via the portal vein and circulatory system where they play a systemic role [33]. Acetate, propionate, and butyrate account for >95% of intestinal SCFA in humans [34,35].

AE promoted an increase in the levels of acetic, propionic, butyric, and isovaleric acids, while mitigating inflammation by regulating the levels of SCFAs. We hypothesized that the anti-inflammatory and antimicrobial properties of AE may exhibit a synergistic relationship, although further investigation is necessary, particularly in the context of intestinal microbiota transplantation in mice. The results of our study suggest that AE exhibits therapeutic properties in the treatment of sepsis. Nevertheless, there remains a scarcity of comprehensive understanding regarding the specific targets, pharmacodynamics, and toxicological facets of AE. Hence, it is recommended that future research endeavors prioritize the clarification of the influence of AE on molecular signal transduction mechanisms and delve deeper into its toxicological implications.

## 4. Materials and Methods

### 4.1. Chemicals and Reagents

AE was purchased from Shanghai Yuanye Biotechnology Co., Ltd. (Shanghai, China). Green qPCR SuperMix reverse transcription and bicinchoninic acid (BCA) protein concentration kits were purchased from TransGen (Beijing, China). Manufacturers of the antibodies used in this study are listed in Table 1.

### 4.2. Cell Lines and Culture Conditions

We examined the anti-inflammatory effects of AE through in vitro experimentation utilizing murine macrophages (RAW 264.7) procured from the American Type Culture Collection (Manassas, VA, USA). The cells were subjected to culture in Dulbecco’s modified Eagle’s medium (Gibco, Grand Island, NY, USA), supplemented with 10% (*v*/*v*) fetal bovine serum and 1% (*v*/*v*) penicillin/streptomycin (Gibco) and maintained in a 5% CO_2_ incubator at 37 °C.

### 4.3. Cell Viability Assay

Cell viability was assessed using Cell Counting Kit-8 (CCK-8; Beyotime Biotechnology, Beijing, China) as previously described [36].

### 4.4. Animals

An equal number of specific-pathogen-free male and female C57BL/6 mice (8–10 weeks old) were procured from Wu’s Animal Centre (Fuzhou, China) and raised at the Experimental Animal Centre of Fujian Normal University. The mice were kept in a room with a temperature range of 23 to 25 °C, humidity between 40 and 60%, and a 12 h light/dark cycle. The mice were provided with ad libitum access to food and water and were grouped into male (20–22 g) and female (18–20 g) groups. Animal experiments were conducted in accordance with the *Guide for the Care and Use of Laboratory Animals* and were approved by the Fujian Provincial Office for Managing Laboratory Animals and the Fujian Normal University Animal Care and Use Committee (approval no. 201800013).

### 4.5. CLP Model of Sepsis

The CLP model was used to induce sepsis as previously described [37]. Mice were anesthetized with an intraperitoneal injection of 0.3% pentobarbital sodium, secured on an operating table, and underwent hair removal and disinfection of the left lower abdomen with a betadine solution and 75% alcohol. Subsequently, a longitudinal incision was made in the appropriate area, followed with sequential incisions of approximately 1 cm in the cortical and muscle layers to expose the caecum. In a meticulous manner, the caecum was extracted using tweezers and secured with a 4.0 sterile surgical line at a distance of 2/3 from the bottom of the ileal valve. A sterile 5 mL syringe needle was used to create a puncture in the ligated region, followed by gentle squeezing of the caecum to expel a small quantity of fecal matter. Subsequently, the ligated and punctured caecum was repositioned within the abdominal cavity, and the muscle and skin layers were sequentially sutured using a 4.0 sterile surgical line. To provide liquid resuscitation, a subcutaneous injection of 1 mL of preheated 0.9% normal saline at 37 °C was administered to the back of the mice, followed by placement on a heating pad until full recovery. Postoperative analgesia was achieved through subcutaneous injection of tramadol (20 mg/kg body weight). The control group underwent the same laparotomy procedure without ligation or puncture.

### 4.6. Experimental Protocol

The mice were divided into six categories in a random manner: Control, CLP model, and different AE treatment groups (0.1, 1.0, and 10 mg/kg/day, n = 10, AE was dissolved in DMSO and diluted with normal saline when administered). These treatments were administered through intraperitoneal injection one hour after inducing sepsis. Survival was monitored at 6 h intervals following drug administration, while sepsis severity was evaluated using MSS [38]. Intraperitoneal injection of pentobarbital sodium salt was used to anesthetize the mice, following which blood, fecal, and tissue samples were collected as previously described [36].

### 4.7. Quantitative Reverse Transcription PCR (RT-qPCR)

Total RNA was isolated using TRIzol (Takara, Tokyo, Japan), and qRT-PCR was conducted as previously described [36]. mRNA amplification was performed using the primers listed in Table 2.

### 4.8. ELISA

ELISA was employed to evaluate the concentrations of IFN-β, IL-1β, IL-6, and TNF-α, according to the manufacturer’s instructions (IFN-β:424001, Thermo Fisher Scientific; IL-6: SM6000B, IL-1β: SMLB00C, TNF-α: SMTA00B, R&D Systems, Minneapolis, MN, USA).

### 4.9. Western Blotting

Western blotting was performed as previously described [39].

### 4.10. Histopathological Examination

H&E staining was performed as previously described [39].

### 4.11. Hematological Parameters

Hematological parameters were assessed by measuring the levels of peripheral blood cells, including red blood cells, lymphocytes, monocytes, granulocytes, platelet hematocrit, and platelets, using an automated animal blood cell analyzer (Mindray Blood cells-2800VET, Mindray Animal Care, Shenzhen, China), as per manufacturer’s guidelines.

### 4.12. Immunohistochemical Assay

Immunohistochemical assays were performed as previously described [40]. Briefly, paraffin was removed from the sections using ethyl alcohol and ultrapure water, and the antigens were unmasked using a microwave oven. Sections were incubated with 3% hydrogen peroxide to impede endogenous enzymes and with blocking solutions to prevent nonspecific antigen binding. Sections were then incubated with an anti-NF-κB (p65) antibody at 4 °C overnight, followed by incubation with a secondary antibody for 1 h. Subsequently, 3,3′-diaminobenzidine tetrahydrochloride hydrate and hematoxylin staining solution were used to stain the sections.

### 4.13. DNA Extraction and 16S rRNA Sequencing

DNA extraction and 16S rRNA sequencing were performed as described [23]. Bioinformatic analysis was conducted using the BMK Cloud platform (https://www.biocloud.net) accessed on 15 November 2022 [41]. 

### 4.14. Detection of SCFAs

SCFAs were extracted and detected as previously described using gas chromatography–mass spectrometry (Shimadzu, Kyoto, Japan) [42].

### 4.15. Statistical Analysis

The obtained images were processed using Adobe Photoshop and Illustrator 2020 software and ImageJ v1.8.0 from the National Institutes of Health. Statistical analyses were conducted using GraphPad Prism (v8.0) software with an unpaired two-tailed *t*-test, one-way ANOVA, two-way ANOVA, or the Mantel–Cox test employed for this purpose. Data are presented as mean ± standard deviation, with statistical significance set at *p* < 0.05.

## 5. Conclusions

In summary, AE administration enhanced the survival rate of mice with CLP-induced sepsis by effectively suppressing the release of pro-inflammatory cytokines and downregulating the NF-κB signaling pathway. AE was found to modulate the diversity and abundance of gut microbiota during sepsis. These findings will help to facilitate the development of AE as a potential therapeutic agent for the management of sepsis. Nonetheless, due to the multifaceted and intricate nature of the anti-inflammatory and antimicrobial properties of AE, further investigation is necessary to explore the synergistic relationship.

## Figures and Tables

**Figure 1 ijms-24-11972-f001:**
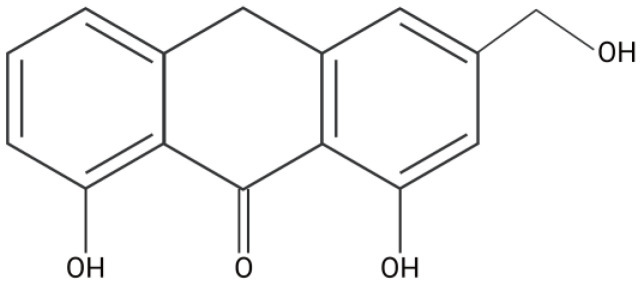
Structures of aloe-emodin (AE).

**Figure 2 ijms-24-11972-f002:**
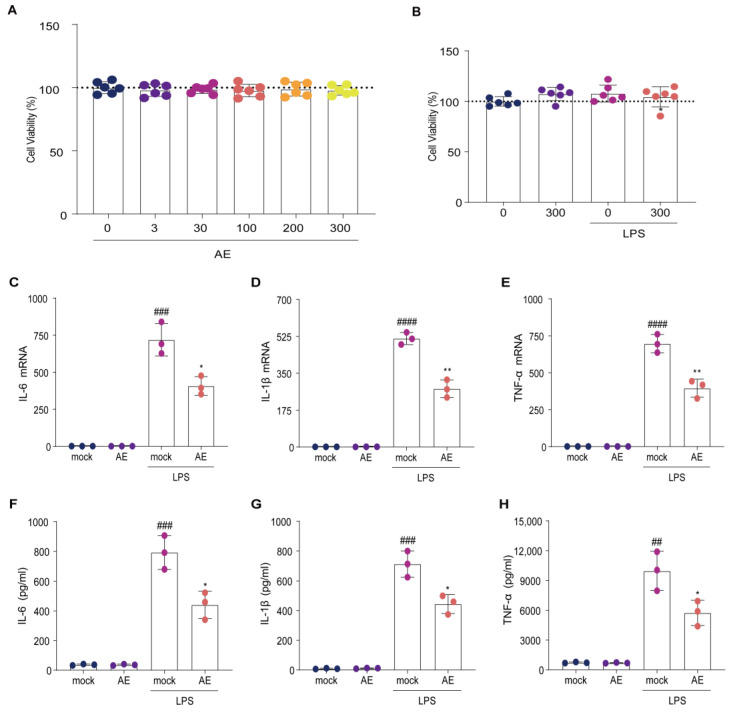
Effects of AE on LPS-stimulated RAW264.7 cells. (**A**) Effect of AE treatment (0–300 μmol/L) on cell viability. (**B**) Effect of AE (300 μmol/L) combined with LPS (0.1 μg/mL) on cell viability. (**C**–**E**) *q*RT-PCR analysis of mRNA levels of the cytokines in RAW264.7 cells following treatment with AE (300 μmol/L) and LPS (0.1 μg/mL) combination; (**C**) *IL-6*; (**D**) *IL-1β*; (**E**) *TNF-α*. (**F**–**H**) ELISA showing changes in protein levels of cytokines in RAW264.7 cells following treatment with AE (300 μmol/L) and LPS (0.1 μg/mL) combination; (**F**) IL-6; (**G**) IL-1β; (**H**) TNF-α. Analysis of variance (ANOVA) and Tukey’s post hoc test were used to analyze the data. ^##^
*p* < 0.01, ^###^
*p* < 0.001, and ^####^
*p* < 0.0001 vs. the control group; * *p* < 0.05 and ** *p* < 0.01 vs. the LPS group. The experiment was repeated three times.

**Figure 3 ijms-24-11972-f003:**
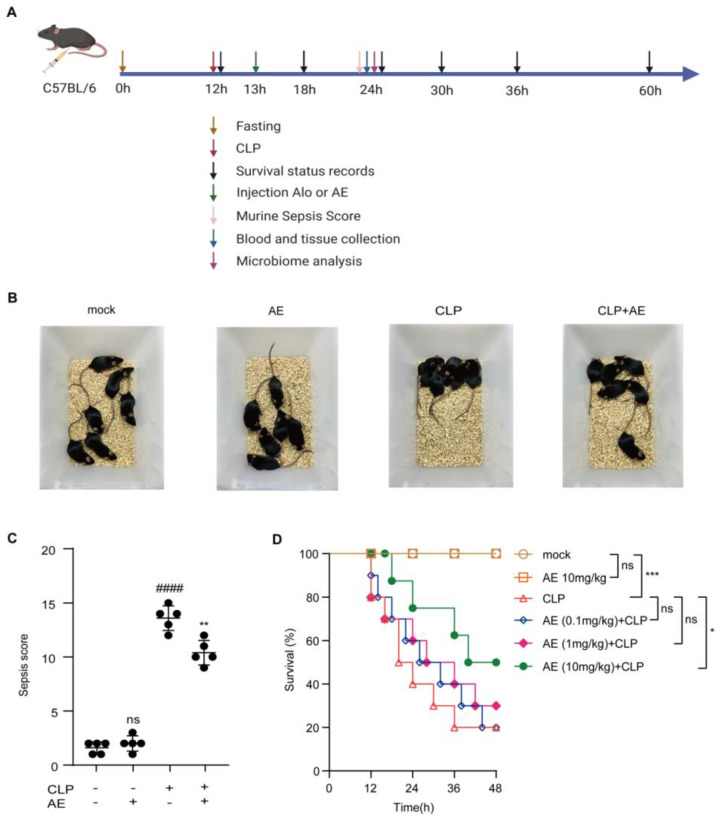
Therapeutic effect of AE in mice with CLP-induced sepsis. (**A**) Experimental design. (**B**) Typical behavioral changes in mice with CLP-induced sepsis after 12 h of AE (10 mg/kg) treatment. (**C**) The murine sepsis score (MSS) in mice with CLP-induced sepsis after 12 h of AE (10 mg/kg) treatment (n = 6); ANOVA and Tukey’s post hoc test were used to analyze the data. ^###*#*^
*p* < 0.0001 vs. the control group; ** *p* < 0.01 vs. the LPS group. (**D**) Effect of AE on the survival of septic mice (n = 20). Data were statistically analyzed using the Mantel–Cox test; * *p* < 0.05 and *** *p* < 0.001; ns, not significant. Sample sizes are indicated in brackets.

**Figure 4 ijms-24-11972-f004:**
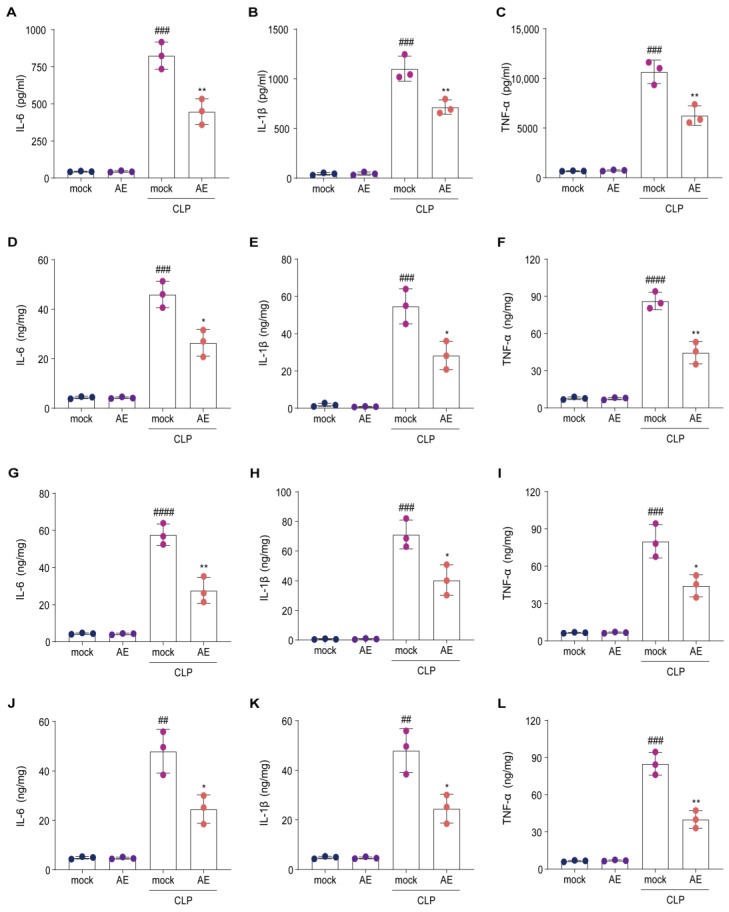
Alterations in inflammatory cytokines in mice afflicted with sepsis induced by CLP subsequent to treatment with AE. The levels of IL-1β, IL-6, and TNF-β in (**A**–**C**) the serum, (**D**–**F**) lung, (**G**–**I**) liver, and (**J**–**L**) heart tissue of mice with sepsis were measured using ELISA, 12 h after AE treatment. ANOVA and Tukey’s post hoc tests were performed to analyze the data. ^#*#*^
*p* < 0.01, ^##*#*^
*p* < 0.001, and ^###*#*^
*p* < 0.0001 vs. the control group; * *p* < 0.05 and ** *p* < 0.01 vs. the CLP group. The experiments were repeated three times.

**Figure 5 ijms-24-11972-f005:**
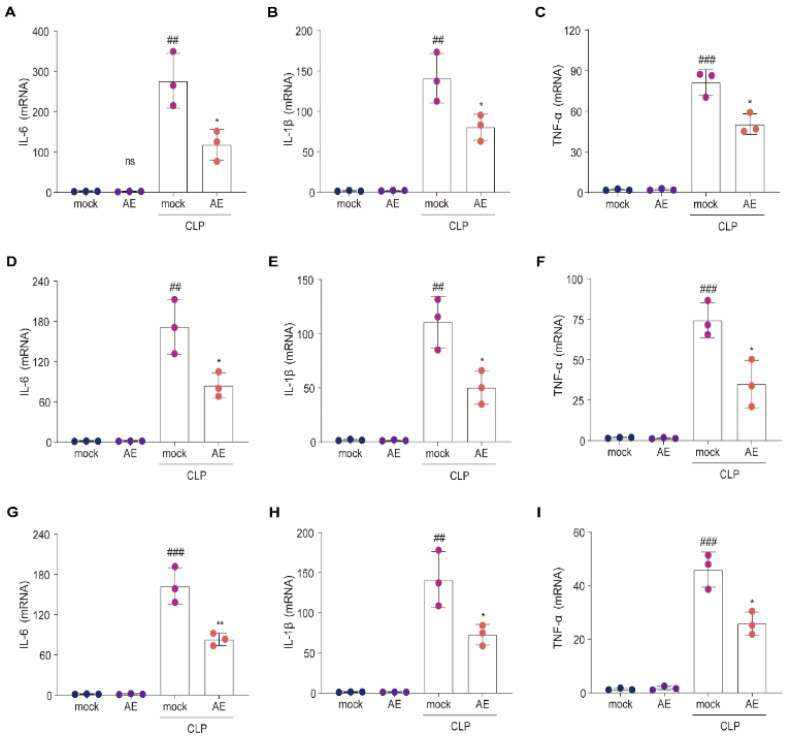
Quantification of pro-inflammatory cytokine levels in mice afflicted with sepsis induced by CLP. The mRNA expression of IL-6, IL-1β, and TNF-α in the (**A**–**C**) lung, (**D**–**F**) liver, and (**G**–**I**) heart tissue of CLP-induced sepsis mice was measured using *q*RT-PCR after 12 h of AE treatment. ANOVA and Tukey’s post hoc tests were performed to analyze the data. ^#*#*^
*p* < 0.01 and ^##*#*^
*p* < 0.001 vs. the control group. * *p* < 0.05 and ** *p* < 0.01 vs. the CLP group. The experiments were repeated three times.

**Figure 6 ijms-24-11972-f006:**
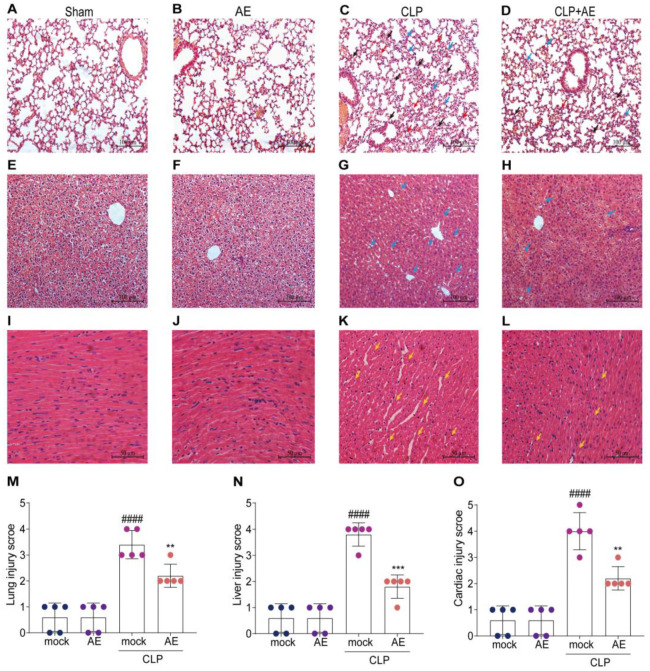
AE’s protective effects against lung, liver, and heart tissue lesions in mice with CLP-induced sepsis. Representative microscopic images of hematoxylin and eosin (H&E) staining of (**A**–**D**) lung, (**E**–**H**) liver, and (**I**–**L**) heart tissue from mice with CLP-induced sepsis (magnification, 200×). Tissue injury is indicated by pulmonary edema (red arrow), inflammatory cell infiltration (blue arrow), alveolar wall thickening (black arrow), and cardiomyocyte rupture (yellow arrow). Pathological scores of (**M**) lung, (**N**) liver, and (**O**) heart tissues of mice with CLP-induced sepsis. Data are expressed as mean ± standard deviation. ^###*#*^
*p* < 0.0001 vs. the control group; ** *p* < 0.01 and *** *p* < 0.001 vs. the CLP group. Scale bars = 50 μm. The experiments were repeated three times.

**Figure 7 ijms-24-11972-f007:**
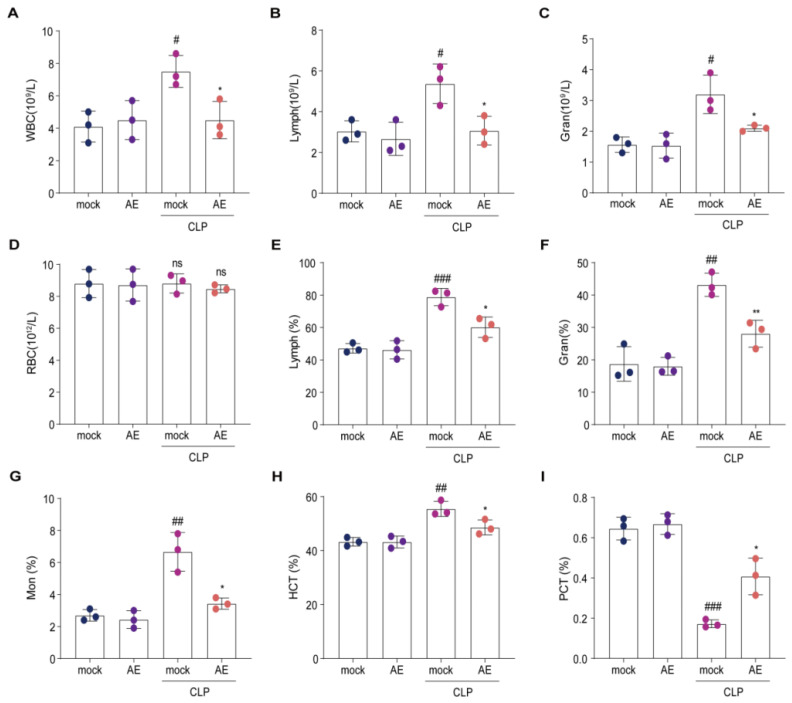
Changes in hematology parameters following AE treatment in mice with CLP-induced sepsis. Count of (**A**) white blood cells (WBC), (**B**) lymphocytes (LymPh), (**C**) granulocytes (Gran), and (**D**) red blood cell (RBC) count and percentage of (**E**) LymPh, (**F**) Gran, and (**G**) monocytes (Mon), (**H**) hematocrit (HCT), and (**I**) platelet crit (PCT) in peripheral blood were analyzed at 12 h after AE treatment in mice with CLP-induced sepsis. ANOVA and Tukey’s post hoc test were performed to analyze the data. *^#^ p* < 0.05, ^#*#*^
*p* < 0.01, and ^##*#*^
*p* < 0.001 vs. the control group. * *p* < 0.05 and ** *p* < 0.01 vs. the CLP group; ns, not significant. The experiments were repeated three times.

**Figure 8 ijms-24-11972-f008:**
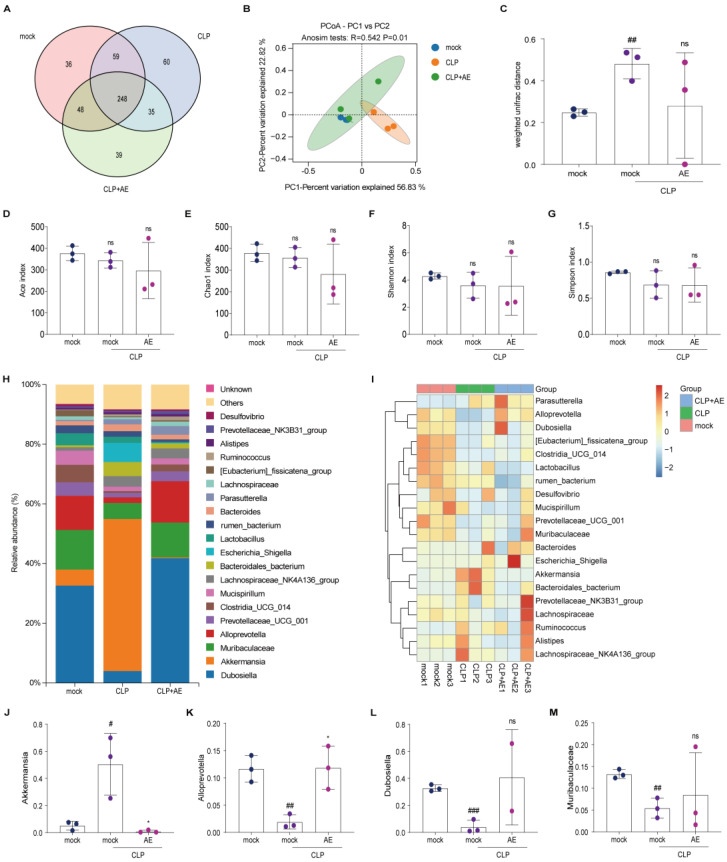
AE restores intestinal microbiota homeostasis in mice with CLP-induced sepsis. (**A**) Operational taxonomic units (OTUs) of intestinal microbes are shown in a Venn diagram. (**B**,**C**) Microbiota diversity analysis at the OTU level of the intestinal microbiota. (**B**) Diverse groups of UniFrac Adonis PCoA plots for beta diversity. (**C**) Weighed UniFrac ANOSIM analysis of beta diversity in distinct populations. (**D**–**G**) OTU-level analysis of intestinal microbiota alpha diversity. (**D**) ACE index; (**E**) Chao1 index; (**F**) Shannon index; (**G**) Simpson index. (**H**) Genus-level composition of gut microbiota. (**I**) Intestinal microorganisms in distinct groups at the genus level are represented by heat maps. Relative abundance of (**J**) *Akkermansia*, (**K**) *Dubosiella*, (**L**) *Muribaculaceae*, (**M**) *Prevotellaceae_UCG_001*. Based on the standardization of the relative abundance of species in each row, the heat map’s Z-value was obtained. ANOVA and Tukey’s post hoc test were performed to analyze the data. *^#^ p* < 0.05, ^#*#*^
*p* < 0.01, and ^##*#*^
*p* < 0.001 vs. the control group. * *p* < 0.05 vs. the CLP group; ns, not significant. The experiments were repeated three times.

**Figure 9 ijms-24-11972-f009:**
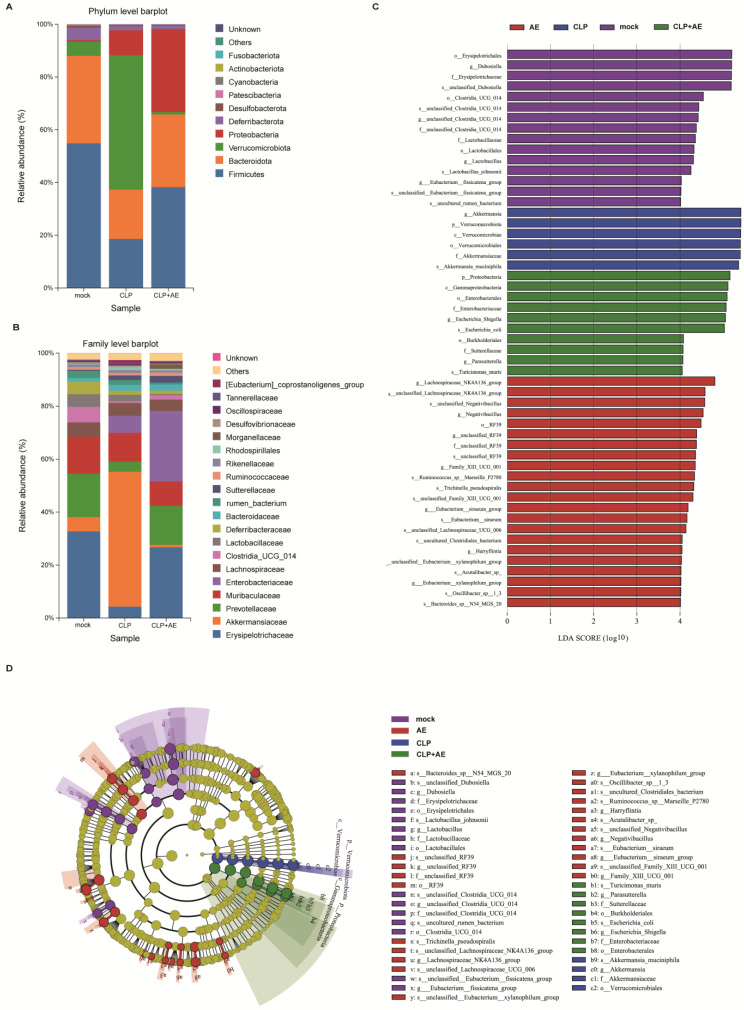
AE treatment alleviates gut microbiota dysbiosis in mice with CLP-induced sepsis. Microbiota composition at the (**A**,**B**) family levels. (**C**) Gut microbiota comparison based on LEfSe (LDA > 4.0). (**D**) LEfSe analysis generated the cladogram.

**Figure 10 ijms-24-11972-f010:**
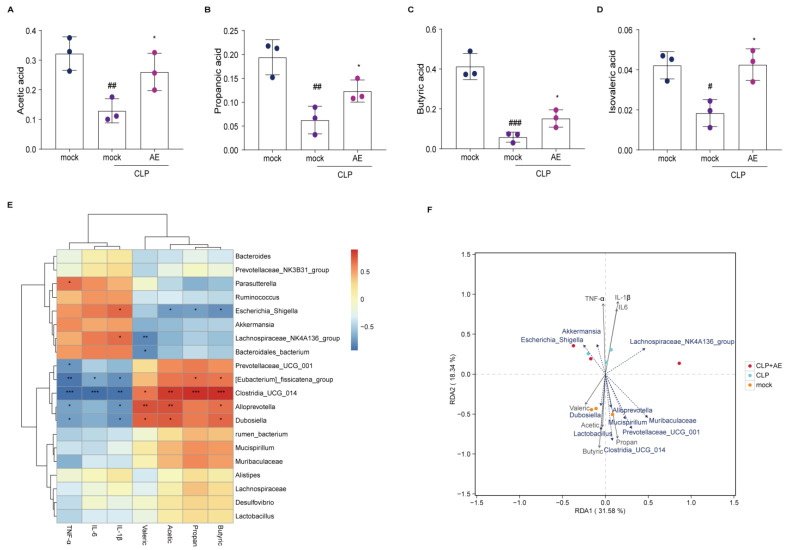
Association between intestinal flora and environmental factors in mice with CLP-induced sepsis following AE treatment. Effects of AE on the levels of short-chain fatty acids (SCFAs): (**A**) acetic, (**B**) propionic, (**C**) butyric, and (**D**) isovaleric acids produced by intestinal microorganisms in CLP-induced septic mice (n = 3). (**E**) Spearman’s rank correlation heat map among bacterial genera, levels of SCFAs, IL-6, IL-1β, and TNF-α in the serum of mice with CLP-induced sepsis. (**F**) Redundancy analysis/canonical correspondence analysis between the bacterial genera and SCFA levels in mice with CLP-induced sepsis. Data are expressed as mean ± standard deviation. ANOVA and Tukey’s post hoc tests were performed to analyze the data. *^#^ p* < 0.05, ^#*#*^
*p* < 0.01, and ^##*#*^
*p* < 0.001 vs. the control group. * *p* < 0.05 vs. the CLP group. The experiments were repeated three times.

**Figure 11 ijms-24-11972-f011:**
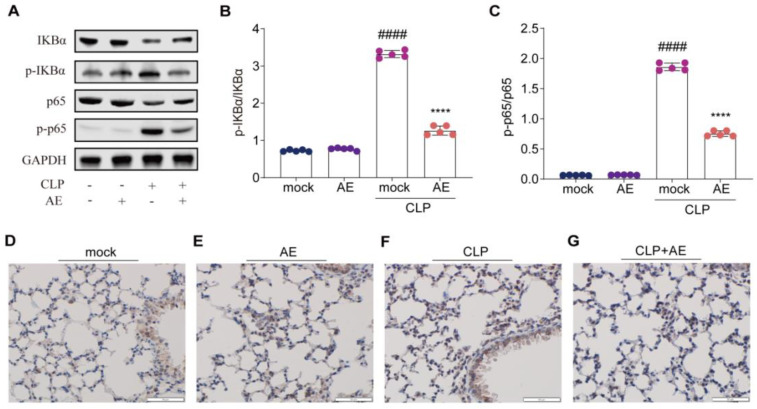
Effect of AE treatment on NF-κB signaling in mice with CLP-induced sepsis. (**A**) Western blot analyses showing the protein expression levels of p-IκBα and p-NF-κB (p-p65) in lung homogenates after 12 h of AE treatment. (**B**,**C**) Image J was used for densitometric analysis of the relevant bands. (**D**–**G**) Immunohistochemistry analysis of lung tissues showing the expression of p-NF-κB (p-p65). Scale bar = 50 μm. ANOVA and Tukey’s post hoc test were performed to analyze the data. ^###*#*^
*p* < 0.0001 vs. the control group. **** *p* < 0.0001 vs. the CLP group. The experiments were repeated three times.

**Table 1 ijms-24-11972-t001:** Antibodies used for the experiments.

Antibody	Source	Identifier
NF-kB p65	ABclonal	A19653
Phospho-NF-kB p65	ABclonal	AP0124
GAPDH	ABclonal	A19056
IκBα	ABclonal	A1187
Phospho-IκBα	ABclonal	AP0999

**Table 2 ijms-24-11972-t002:** Sequence of the primers.

Name	Sequence (5′–3′)
IL-6	F: TACCACTTCACAAGTCGGAGGC
	R: CTGCAAGTGCATCATCGTTGTTC
IL-1β	F: TGGGAAACAACAGTGGTCAGG
	R: CCATCAGAGGCAAGGAGGAA
TNF-α	F: GAGTGACAAGCCTGTAGCC
	R: CTCCTGGTATGAGATAGCAAA

IL, interleukin; TNF, tumor necrosis factor; F, forward; R, reverse.

## Data Availability

All data included in this study are available upon request by contact with the corresponding author.

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
