# Peer review of "Aloe-Emodin Ameliorates Cecal Ligation and Puncture-Induced Sepsis"

_ijms, 2023, doi:10.3390/ijms241511972_

Round 1

Reviewer 1 Report

This article addresses a subject of interest: the research for new substances with promising results in treating sepsis, a life-threatening condition that requires a rapid and efficient approach.

The paper deserves to be published in IJMS, however, there are several points in which improvements should be made to strengthen the manuscript.

·       Line 33 mentions sepsis – 3, please briefly explain this notion and provide the appropriate references.

·       Figure 2 – line 93 – Please replace C, D, E with F, G, H.

·       Figure 3A mentions aloin or aloe-emodin, this can lead to the expectation that both substances have been studied, but the Result and Materials and Methods only refer to aloe - emodin. Please clarify.   

·       Official regulations regarding the interdiction of aloe-emodin, due to toxicity concerns, have been enforced throughout the European Union. I consider they should be mentioned. Also, the difference between a drug and a food supplement can be outlined.

·       Please describe the formulation of the injected AE solution and clearly mention the rout of administration. The term injection creates the expectation of a parenteral route.

·       Please mention the number of animals used for each experiment in the Materials and Methods section.

·       A paragraph regarding the limitations of the study would be beneficial to the reader.

·       The perspectives of the study should be better outlined.

·       References 15 and 21 are identical, please revise.

Author Response

July 18, 2023

Prof. Dr. Maria Luz Fernandez

Editor-in-Chief

Ms. Luna Liu

Assistant Editor

Dear Editor,

I wish to re-submit the manuscript titled “Aloe-emodin ameliorates cecal ligation and puncture-induced sepsis.” The manuscript ID is ijms-2496527.

We express our gratitude to you and the reviewers for your valuable suggestions and insightful perspectives. The manuscript has greatly benefited from these astute recommendations.

Attached is the revised version of our manuscript. In the following pages are our point-by-point responses to each of the comments of the reviewers. Revisions in the text are highlighted by the utilization of the color red. We hope that the revisions in the manuscript and our accompanying responses would be sufficient to make our manuscript suitable for publication in International Journal of Molecular Sciences.

Thank you for your consideration. I look forward to hearing from you.

Sincerely,

Jingqian Su, Ph.D.

Associate Professor

Fujian Key Laboratory of Innate Immune Biology,

Biomedical Research Center of South China,

College of Life Science,

Fujian Normal University, Fuzhou 350117, Fujian, China

Tel: +86-18950498937

E-mail: sjq027@fjnu.edu.cn

Qi Chen, Ph.D.

Professor

Fujian Key Laboratory of Innate Immune Biology

Biomedical Research Center of South China

College of Life Science, Fujian Normal University

Fuzhou 350117, China

chenqi@fjnu.edu.cn

Duo Chen, Ph.D.

The Public Service Platform for Industrialization Development Technology of Marine Biological Medicine and Products of the State Oceanic Administration,

Fujian Key Laboratory of Special Marine Bioresource Sustainable Utilization,

Southern Institute of Oceanography,

College of Life Science,

Fujian Normal University, Fuzhou 350117, Fujian, China

E-mail: chenduo@fjnu.edu.cn

Responses to the comments of Reviewer #1

  1. Line 33 mentions sepsis – 3, please briefly explain this notion and provide the appropriate references.

Response:

Many thanks for the reviewer suggestion. We are grateful for the suggestion.

(1)In the revised manuscript (line 33-38), the content of sepsis-3 has been incorporated, as follow:

Currently, the most recent scholarly investigation on sepsis is denoted as "Sepsis 3.0". Within the framework of "Sepsis 3.0", sepsis is delineated as a perilous impairment of organ function, arising from an aberrant host reaction to infection. This implies that the body's defensive response to infection detrimentally affects its own tissues and organs, thereby posing a threat to life [2].

(2)The citation has been included at line 515.

  1. Figure 2 – line 93 – Please replace C, D, E with F, G, H.

Response:

Thank you for your comment and we apologize for the confusion. The graphic annotation has undergone revision at line 107.

  1. Figure 3A mentions aloin or aloe-emodin, this can lead to the expectation that both substances have been studied, but the Result and Materials and Methods only refer to aloe - emodin. Please clarify.

Response:

We express our sincere gratitude for the valuable comment provided. Aloin has the potential to undergo hydrolysis, resulting in the formation of aloe emodin. To accurately depict the structural modifications, Figure 1 illustrates the structural formulas of both compounds. However, the Results, Materials and Methods sections solely focus on aloe-emodin, disregarding Aloin. We concur with the reviewer's suggestion that including a comparison between these compounds would yield beneficial results. Consequently, Aloin has been omitted from Figure 1.

  1. Official regulations regarding the interdiction of aloe-emodin, due to toxicity concerns, have been enforced throughout the European Union. I consider they should be mentioned. Also, the difference between a drug and a food supplement can be outlined.

Response:

We express our sincere gratitude for your valuable comment. Supplementary explanations have been provided in line 66-81, as follows:

Over the past decade, numerous reports have provided evidence of the hepatotoxic and nephrotoxic properties of AE [12]. Currently, the European Union has imposed a ban on the utilization of AE due to the potential genotoxicity of its derivatives [13]. Nevertheless, the results obtained from an in vivo comet test have unequivocally established that aloe-emodin does not exhibit genotoxic characteristics [14]. Hence, the precise clinical implications of these toxic effects remain uncertain [15]. In contrast to food, AE, being a potential therapeutic agent, should be permitted to induce specific adverse reactions in the human body. However, it is imperative to conduct further investigations into the pharmacological side effects of AE.

In the intestinal tract, Alo hydrolyzes one glucose molecule into AE, which stimulates the intestinal wall, enhances intestinal peristalsis, and promotes the expulsion of intestinal waste [16]. Hu et al. provided evidence that AE exerts anti-inflammatory effects by inhibiting the production of inflammatory factors in RAW264.7 macrophages induced by LPS [17]. Similarly, Gao et al.  demonstrated the protective effects of AE against LPS-induced inflammation in a mouse model. These findings suggest AE as a viable therapeutic agent against sepsis [18].

  1. Please describe the formulation of the injected AE solution and clearly mention the rout of administration. The term injection creates the expectation of a parenteral route.

Response:

We would like to extend our heartfelt appreciation for your invaluable comment. The inclusion of the formulation and administration route of AE solution is documented in lines 416-419, as follow:

The mice were divided into six categories in a random manner: Control, CLP model, and different AE treatment groups (0.1, 1.0, and 10 mg/kg/day, n=10, AE was dissolved in DMSO and diluted with normal saline when administered). These treatments were administered through intraperitoneal injection one hour after inducing sepsis

  1. Please mention the number of animals used for each experiment in the Materials and Methods section.

Response:

We express our gratitude for your inquiry. As shown in line 417 of the revised manuscript, each experimental group consisted of 10 mice.

  1. A paragraph regarding the limitations of the study would be beneficial to the reader.

Response:

I express my gratitude for your valuable suggestions. I have derived substantial benefits from your input and subsequently incorporated the discussion on the limitations of the study within lines 349-354, as presented below:

The results of our study suggest that AE exhibits therapeutic properties in the treatment of sepsis. Nevertheless, there remains a scarcity of comprehensive understanding regarding the specific targets, pharmacodynamics, and toxicological facets of AE. Hence, it is recommended that future research endeavors prioritize the clarification of the influence of AE on molecular signal transduction mechanisms and delve deeper into its toxicological implications.

  1. The perspectives of the study should be better outlined.

Response:

We express our sincere gratitude for the valuable suggestion provided by the reviewer. Additional explanations have been provided in lines 86-90, as follows:

This study involved the implementation of postoperative AE therapy in a CLP-induced sepsis model. The principal objective was to explore the anti-inflammatory impact, assess modifications in immune cells, and scrutinize the composition of intestinal flora. The outcomes of this research are expected to contribute to the advancement of AE as a potential therapeutic approach for sepsis.

  1. References 15 and 21 are identical, please revise.

Response:

We express our gratitude for your valuable comment and extend our sincere apologies for any confusion that may have arisen. The references in line 550 has been revised.

Reviewer 2 Report

This article related to sepsis using the genus Aloe demonstrates its health benefits, including anti-inflammatory properties, inhibits inflammatory responses in mice induced by lipopolysaccharides, indicating its potential as a therapeutic approach for the treatment of sepsis while the implications toxicological aspects of aloe-emodin (AE), extracted from various Aloe species, remain uncertain in clinical contexts, even if in my opinion the toxicological aspect should have been clarified in this working context. However, the article is in line with the aim of the title and the quality of the study is very satisfactory.

Author Response

July 18, 2023

Prof. Dr. Maria Luz Fernandez

Editor-in-Chief

Ms. Luna Liu

Assistant Editor

Dear Editor,

I wish to re-submit the manuscript titled “Aloe-emodin ameliorates cecal ligation and puncture-induced sepsis.” The manuscript ID is ijms-2496527.

We express our gratitude to you and the reviewers for your valuable suggestions and insightful perspectives. The manuscript has greatly benefited from these astute recommendations.

Attached is the revised version of our manuscript. In the following pages are our point-by-point responses to each of the comments of the reviewers. Revisions in the text are highlighted by the utilization of the color red. We hope that the revisions in the manuscript and our accompanying responses would be sufficient to make our manuscript suitable for publication in International Journal of Molecular Sciences.

Thank you for your consideration. I look forward to hearing from you.

Sincerely,

Jingqian Su, Ph.D.

Associate Professor

Fujian Key Laboratory of Innate Immune Biology,

Biomedical Research Center of South China,

College of Life Science,

Fujian Normal University, Fuzhou 350117, Fujian, China

Tel: +86-18950498937

E-mail: sjq027@fjnu.edu.cn

Qi Chen, Ph.D.

Professor

Fujian Key Laboratory of Innate Immune Biology

Biomedical Research Center of South China

College of Life Science, Fujian Normal University

Fuzhou 350117, China

chenqi@fjnu.edu.cn

Duo Chen, Ph.D.

The Public Service Platform for Industrialization Development Technology of Marine Biological Medicine and Products of the State Oceanic Administration,

Fujian Key Laboratory of Special Marine Bioresource Sustainable Utilization,

Southern Institute of Oceanography,

College of Life Science,

Fujian Normal University, Fuzhou 350117, Fujian, China

E-mail: chenduo@fjnu.edu.cn

Responses to the comments of Reviewer #2

  1. This article related to sepsis using the genus Aloe demonstrates its health benefits, including anti-inflammatory properties, inhibits inflammatory responses in mice induced by lipopolysaccharides, indicating its potential as a therapeutic approach for the treatment of sepsis while the implications toxicological aspects of aloe-emodin (AE), extracted from various Aloe species, remain uncertain in clinical contexts, even if in my opinion the toxicological aspect should have been clarified in this working context. However, the article is in line with the aim of the title and the quality of the study is very satisfactory.

Response:

We extend our utmost appreciation for the comment and convey our sincere gratitude for the invaluable suggestion put forth by the reviewer. The manuscript has provided substantial clarification regarding the advancements made in the domain of AE toxicology (lines 66-74), as follows:

Over the past decade, numerous reports have provided evidence of the hepatotoxic and nephrotoxic properties of AE [12]. Currently, the European Union has imposed a ban on the utilization of AE due to the potential genotoxicity of its derivatives [13]. Nevertheless, the results obtained from an in vivo comet test have unequivocally established that aloe-emodin does not exhibit genotoxic characteristics [14]. Hence, the precise clinical implications of these toxic effects remain uncertain [15]. In contrast to food, AE, being a potential therapeutic agent, should be permitted to induce specific adverse reactions in the human body. However, it is imperative to conduct further investigations into the pharmacological side effects of AE.
